# EVALUATING THE ROBUSTNESS OF TEXT-TO-IMAGE DIFFUSION MODELS AGAINST REAL-WORLD ATTACKS

## ABSTRACT

Text-to-image (T2I) diffusion models (DMs) have shown promise in generating high-quality images from textual descriptions. The real-world applications of these models require particular attention to their safety and fidelity, but this has not been sufficiently explored. One fundamental question is whether the existing T2I DMs are robust against variations over input texts. To answer it, this work provides the first robustness evaluation of T2I DMs against *real-world* perturbations. Unlike malicious attacks that involve apocryphal alterations to the input texts, we consider a perturbation space spanned by realistic errors (e.g., typo, glyph, phonetic) that humans can make and adopt adversarial attacks to generate worst-case perturbations for robustness evaluation. Given the inherent randomness of the generation process, we develop novel distribution-based objectives to mislead T2I DMs. We optimize the objectives by black-box attacks without any knowledge of the model. Extensive experiments demonstrate the effectiveness of our method for attacking popular T2I DMs and simultaneously reveal their non-trivial robustness issues. Moreover, we provide an in-depth analysis of our method to show that it is not designed to attack the text encoder in T2I DMs solely.

## 1 INTRODUCTION

Diffusion models (DMs) (Sohl-Dickstein et al., 2015; Ho et al., 2020) have demonstrated remarkable success in generating images and shown promise in diverse applications, including super-resolution (Saharia et al., 2022b), image inpainting (Lugmayr et al., 2022), text-to-image synthesis (Rombach et al., 2022; Ramesh et al., 2022), video generation (Ho et al., 2022a;b), etc. A typical DM employs a forward process that gradually diffuses the data distribution towards a noise distribution and a reverse process that recovers the data through step-by-step denoising. Among the applications, text-to-image (T2I) generation has received significant attention and witnessed the development of large models such as GLIDE (Nichol et al., 2022), Imagen (Saharia et al., 2022a), DALL-E 2 (Ramesh et al., 2022), Stable Diffusion (Rombach et al., 2022), VQ-Diffusion (Gu et al., 2022), etc. These models typically proceed by conditioning the reverse process on the embeddings of textual descriptions obtained from certain text encoders. Their ability to generate high-quality images from textual descriptions can significantly simplify the creation of game scenarios, book illustrations, organization logos, and more.

However, T2I DMs are shown to be vulnerable to adversarial attacks (Du et al., 2023). By applying subtle perturbations to the input text, the generated image deviates significantly from the intended target. Hence, evaluating the robustness of T2I DMs is a fundamental problem in this regard, often achieved by adversarial attacks. Some initial studies in this field have demonstrated that manipulating the input text by creating meaningless or distorted custom words (Millière, 2022) or phrases (Maus et al., 2023), or adding irrelevant distractions (Zhuang et al., 2023) can lead to significant bias in the semantics of the images generated by T2I DMs. However, it should be noted that these works primarily focus on malicious attacks, which often introduce substantial changes to the text and may rarely occur in real-world scenarios. To bridge this gap, we shift our attention from intentional attacks to everyday errors such as typos, grammar mistakes, or vague expressions, as suggested by related works in natural language processing (Li et al., 2018; Eger & Benz, 2020; Eger et al., 2019a; Le et al., 2022), to thoroughly evaluate the robustness of models that interact with humans in **pratical** use.

This work provides the first evaluation of the robustness of T2I DMs against *real-world* perturbations. As discussed, we consider an attack space spanned by realistic errors that humans can make to ensure

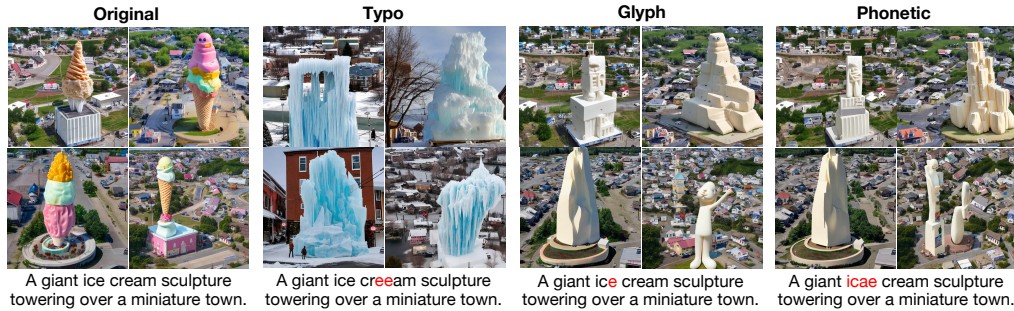

| Original | Typo | Glyph | Phonetic |
|---|---|---|---|

A giant ice cream sculpture towering over a miniature town.  A giant ice creeam sculpture towering over a miniature town.  A giant ice cream sculpture towering over a miniature town.  A giant icae cream sculpture towering over a miniature town.

Figure 1: An illustration of our attack method against Stable Diffusion (Rombach et al., 2022) based on three attack rules (detailed in Section 3.3.1). Adversarially modified content is highlighted in red. Note that the red 'e'(U+0435) in Glyph is different from 'e'(U+0065) in the original sentence.

semantic consistency, including typos, glyphs, and phonetics. To tackle the inherent uncertainty in the generation process of DMs, we develop novel distribution-based attack objectives to mislead T2I DMs. We perform attacks in a black-box manner using greedy search to avoid assumptions about the model. Technically, our attack algorithm first identifies the keywords based on the words' marginal influence on the generation distribution and then applies elaborate character-level replacements. Our algorithm can be used by the model developers to evaluate the robustness of their T2I models before being deployed in the wild.

We perform extensive empirical evaluations on datasets of artificial prompts and image captions. We first conduct a set of diagnostic experiments to prioritize the different variants originated from the distribution-oriented attack objectives, which also reflects the vulnerability of existing T2I DMs. We then provide an interesting discussion on the target of attacking DMs: the text encoder only vs. the whole diffusion process. Finally, we attack T2I DMs (including DALL-E 2) in real-world settings and observe high success rates, even in the case that the perturbation rates and query times are low.

## 2 RELATED WORK

**Diffusion models** (DMs) (Sohl-Dickstein et al., 2015; Ho et al., 2020; Song et al., 2020) are a poweful family of generative models that attract great attention recently. In the diffusion process, the data distribution is diffused to an isotropic Gaussian by continually adding Gaussian noises. The reverse process recovers the original input from a Gaussian noise by denoising. DMs have been widely applied to text-to-image (T2I) generation. GLIDE (Nichol et al., 2022) first achieves this by integrating the text feature into transformer blocks in the denoising process. Subsequently, increasing effort is devoted to this field to improve the performance of T2I generation, with DALL-E (Ramesh et al., 2021), Cogview (Ding et al., 2021), Make-A-Scene (Gafni et al., 2022), Stable Diffusion (Rombach et al., 2022), and Imagen (Saharia et al., 2022a) as popular examples. A prevalent strategy nowadays is to perform denoising in the feature space while introducing the text condition by cross-attention mechanisms (Tang et al., 2022). However, textual conditions cannot provide the synthesis results with more structural guidance. To remediate this, there are many other kinds of DMs conditioning on factors beyond text descriptions, such as PITI (Wang et al., 2022a), ControlNet (Zhang & Agrawala, 2023) and Sketch-Guided models (Voynov et al., 2022).

**Adversarial attacks** typically deceive DNNs by integrating carefully-crafted tiny perturbations into input data (Szegedy et al., 2014; Zhang et al., 2020). Based on how an adversary interacts with the victim model, adversarial attacks can be categorized into white-box attacks (Zhang et al., 2022a; Meng & Wattenhofer, 2020; Xu et al., 2022) (with full access to the victim model) and black-box attacks (Zhang et al., 2022b; He et al., 2021) (with limited access to the victim model). Adversarial attacks on text can also be categorized in terms of the level of granularity of the perturbations. Character-level attacks (Eger et al., 2019b; Formento et al., 2023) modify individual characters in words to force the tokenizer to process multiple unrelated embeddings instead of the original, resulting in decreased performance. Word-level attacks (Li et al., 2021; Lee et al., 2022) employ a search algorithm to locate useful perturbing embeddings or operations that are clustered close to the candidate attack word's embedding given a similarity constraint (e.g., the Universal Sentence Encoder (Cer et al., 2018)). Sentence-level attacks (Wang et al., 2020; Han et al., 2020) refer to making changes to sentence structures in order to prevent the model from correctly predicting the outcome. Multi-level attacks

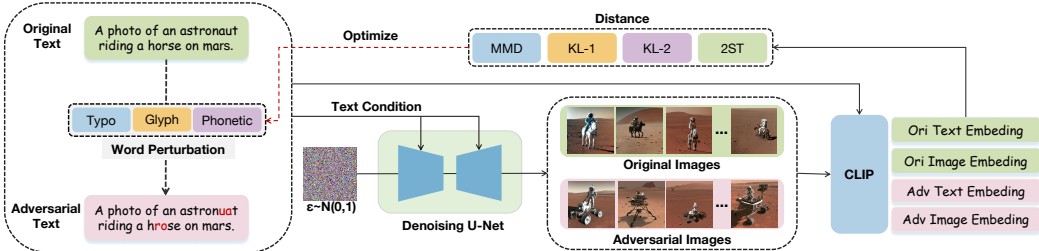

Figure 2: An illustration of our proposed attack method for evaluating the robustness of T2I DMs.

(Gupta et al., 2021; Wallace et al., 2019) combine multiple types of perturbations, making the attack cumulative. Recent studies (Millière, 2022; Maus et al., 2023; Zhuang et al., 2023) have explored the over-sensitivity of T2I DMs to prompt perturbations in the text domain with malicious word synthesis, phrase synthesis, and adding distraction. Zhuang et al. (2023) also reveal the vulnerability of T2I models and attributes it to the weak robustness of the used text encoders.

## 3 METHODOLOGY

This section provides a detailed description of our approach to real-world adversarial attacks of T2I DMs, as shown in Figure 2. We briefly outline the problem formulation before delving into the design of attack objective functions and then describe how to perform optimization in a black-box manner.

### 3.1 PROBLEM FORMULATION

A T2I DM that accepts a text input $c$ and generates an image $x$ essentially characterizes the conditional distribution $p_\theta(x|c)$ with $\theta$ as model parameters. To evaluate the robustness of modern DMs so as to govern their behaviors when adopted in the wild, we opt to attack the input text, i.e., finding a text $c'$ which keeps close to the original text $c$ but can lead to a significantly biased generated distribution. Such an attack is meaningful in the sense of encompassing real-world perturbations such as typos, glyphs, and phonetics. Concretely, the optimization problem is formulated as:

$$\max_{c'} \mathcal{D}(p_\theta(x|c')\|p_\theta(x|c)), \quad \text{s.t. } d(c, c') \leq \epsilon, \tag{1}$$

where $\mathcal{D}$ denotes a divergence measure between two distributions, $d(c, c')$ measures the distance between two texts, and $\epsilon$ indicates the perturbation budget.

The main challenge of attack lies in that we cannot write down the exact formulation of $p_\theta(x|c)$ and $p_\theta(x|c')$ of DMs but get only a few i.i.d. samples $\{\bar{x}_1, \ldots, \bar{x}_N\}$ and $\{x_1, \ldots, x_N\}$ from them, where $\bar{x}_i$ is an image generated with the original text $c$ while $x_i$ is generated with the modified text $c'$.

### 3.2 ATTACK OBJECTIVES

Here, we develop four instantiations of the distribution-based attack objective defined in Eq. (1).

#### 3.2.1 MMD DISTANCE

As validated by the community (Dziugaite et al., 2015; Tolstikhin et al., 2016), the maximum mean discrepancy (MMD) is a widely used metric to distinguish two distributions given finite samples. Formally, assuming access to a kernel function $\kappa$, the square of MMD distance is typically defined as:

$$\mathcal{D}_{\text{MMD}^2}(p_\theta(x|c')\|p_\theta(x|c)) \approx \frac{1}{N^2}\sum_{i=1}^{N}\sum_{j=1}^{N}\kappa(x_i, x_j) - \frac{2}{N^2}\sum_{i=1}^{N}\sum_{j=1}^{N}\kappa(x_i, \bar{x}_j) + C, \tag{2}$$

where $C$ refers to a constant agnostic to $c'$. The feature maps associated with the kernel should be able to help construct useful statistics of the sample set such that MMD can compare distributions. In the case that $x$ is an image, a valid choice is a deep kernel built upon a pre-trained NN-based image encoder $h$ (e.g., a ViT trained by the objective of MAE (He et al., 2022) or CLIP (Radford et al., 2021)). In practice, we specify the kernel with a simple cosine form $\kappa(x, x') := h(x)^\top h(x')/\|h(x)\|\|h(x')\|$ given that $h$'s outputs usually locate in a well-suited Euclidean space.

### 3.2.2 KL DIVERGENCE

Considering that text also provides crucial information in the attack process, we will incorporate text information to consider the joint distribution of images and texts. Due to the excellent ability of CLIP to represent both image and text information while preserving their relationships, we have chosen to use CLIP as the model for encoding images and texts. Assume access to a pre-trained $\phi$-parameterized CLIP model comprised of an image encoder $h_\phi$ and a text encoder $g_\phi$ and assume the output features to be $L^2$-normalized. It can provide a third-party characterization of the joint distribution between the image $x$ and the text $c$ for guiding attack. Note that $h_\phi(x)^\top g_\phi(c)$ measures the likelihood of the coexistence of image $x$ and text $c$, thus from a probabilistic viewpoint, we can think of $e_\phi(x, c) := \alpha h_\phi(x)^\top g_\phi(c)$, where $\alpha$ is some constant scaling factor, as $\log p_\phi(x, c)$. Under the mild assumption that $p_\phi(x|c)$ approximates $p_\theta(x|c)$, we instantiate the measure $\mathcal{D}$ in Eq. (1) with KL divergence and derive the following maximization objective (details are deferred to Appendix):

$$\mathcal{D}_{\text{KL}}(p_\theta(x|c')\|p_\theta(x|c)) \approx \mathbb{E}_{p_\theta(x|c')}[-e_\phi(x, c)] + \mathbb{E}_{p_\theta(x|c')}[\log p_\theta(x|c')] + C, \tag{3}$$

where $C$ denotes a constant agnostic to $c'$. The first term corresponds to generating images containing semantics contradictory to text $c$ and can be easily computed by Monte Carlo (MC) estimation. The second term is negative entropy, so the maximization of it means reducing generation diversity. Whereas, in practice, the entropy of distribution over high-dimensional images cannot be trivially estimated given a few samples. To address this issue, we replace $\mathbb{E}_{p_\theta(x|c')}[\log p_\theta(x|c')]$ with a lower bound $\mathbb{E}_{p_\theta(x|c')}[\log q(x)]$ for any probability distribution $q$, due to that $\mathcal{D}_{\text{KL}}(p_\theta(x|c')\|q(x)) = \mathbb{E}_{p_\theta(x|c')}[\log p_\theta(x|c') - \log q(x)] \geq 0$. In practice, we can only acquire distributions associated with the CLIP model, so we primarily explore the following two strategies.

**Strategy 1.** $\log q(x) := \log p_\phi(x, c') = e_\phi(x, c')$. Combining with Eq. (3), there is ($C$ is omitted):

$$\mathcal{D}_{\text{KL}}(p_\theta(x|c')\|p_\phi(x|c)) \geq \mathbb{E}_{p_\theta(x|c')}[e_\phi(x, c') - e_\phi(x, c)] \approx \alpha\Big[\frac{1}{N}\sum_{i=1}^{N} h_\phi(x_i)\Big]^\top \Big(g_\phi(c') - g_\phi(c)\Big). \tag{4}$$

The adversarial text $c'$ would affect both the generated images $x_i$ and the text embeddings $g_\phi(c')$. Therefore, it is likely that by maximizing the resulting term in Eq. (4) w.r.t. $c'$, the text encoder of the CLIP model is attacked (i.e., $g_\phi(c') - g_\phi(c)$ is pushed to align with the average image embedding), which deviates from our goal of delivering a biased generation distribution.

**Strategy 2.** $\log q(x) := \log p_\phi(x) = \text{L}_{\hat{c} \in \mathcal{C}}(e_\phi(x, \hat{c})) - \log |\mathcal{C}|$ where L is the log-sum-exp operator and $\mathcal{C}$ denotes the set of all possible text inputs. Likewise, there is (we omit constants):

$$\mathcal{D}_{\text{KL}}(p_\theta(x|c')\|p_\phi(x|c)) \geq \mathbb{E}_{p_\theta(x|c')}[\text{L}_{\hat{c} \in \mathcal{C}}(e_\phi(x, \hat{c})) - e_\phi(x, c)] \approx \frac{1}{N}\sum_{i=1}^{N}\big[\text{L}_{\hat{c} \in \mathcal{C}}(e_\phi(x_i, \hat{c})) - e_\phi(x_i, c)\big]. \tag{5}$$

As shown, the first term pushes the generated images toward the high-energy regions, and the second term hinders the generated images from containing semantics about $c$. To reduce the computational overhead, we draw a set of commonly used texts and pre-compute their text embeddings via CLIP before attacking. Then, during attacking, we only need to send the embeddings of generated images to a linear transformation followed by an L operator to get an estimation of the first term of Eq. (5).

### 3.2.3 TWO-SAMPLE TEST

In essence, distinguishing $p_\theta(x|c')$ and $p_\theta(x|c)$ by finite observations corresponds to a two-sample test (2ST) in statistics, and the aforementioned MMD distance is a test statistic that gains particular attention in the machine learning community. Based on this point, we are then interested in building a general framework that can embrace existing off-the-shelf two-sample test tools for attacking T2I DMs. This can considerably enrich the modeling space. Basically, we define a unified test statistic in the following formula

$$\hat{t}\Big(\{\varphi(x_i)\}_{i=1}^{N}, \{\varphi(\bar{x}_i)\}_{i=1}^{N}\Big). \tag{6}$$

Roughly speaking, we will reject the null hypothesis $p_\theta(x|c') = p_\theta(x|c)$ when the statistic is large to a certain extent. The function $\hat{t}$ in the above equation is customized by off-the-shelf two-sample test tools such as KS test, t-test, etc. Considering the behavior of these tools may quickly deteriorate as the dimension increases (Gretton et al., 2012), we introduce a projector $\varphi$ to produce one-dimensional

representations of images $x$. As a result, $\varphi$ implicitly determines the direction of our attack. For example, if we define $\varphi$ as a measurement of image quality in terms of FID (Heusel et al., 2017), then by maximizing Eq. (6), we will discover $c'$ that leads to generations of low quality. Recalling that our original goal is a distribution of high-quality images deviated from $p_\theta(x|c)$, we hence want to set $\varphi(\cdot) := \log p_\theta(\cdot|c)$, which, yet, is inaccessible. Reusing the assumption that the conditional distribution captured by a CLIP model can form a reasonable approximation to $p_\theta(x|c)$, we set $\varphi(\cdot)$ to the aforementioned energy score $e_\phi(\cdot, c)$, which leads to the following test statistic:

$$\mathcal{D}_{2\text{ST}}(p_\theta(x|c')\|p_\theta(x|c)) := \hat{t}\Big(\{e_\phi(x_i, c)\}_{i=1}^N, \{e_\phi(\bar{x}_i, c)\}_{i=1}^N\Big). \quad (7)$$

We empirically found that this t-test can lead to a superior attack success rate over other two-sample test tools, so we use the t-test as the default selection in the following.

## 3.3 ATTACK METHOD

Based on the attack objectives specified above, we define a real-world-oriented search space and employ a greedy search strategy to find adversarial input text for T2I DMs.

### 3.3.1 PERTURBATION RULES

Following related works in natural language processing (Eger & Benz, 2020; Eger et al., 2019a; Le et al., 2022; Chen et al., 2022; 2023), we include the following three kinds of perturbations into the search space of our attack algorithm: (1) **Typo** (Li et al., 2018; Eger & Benz, 2020), which comprises seven fundamental operations for introducing typos into the text, including randomly deleting, inserting, replacing, swapping, adding space, transforming case, and repeating a single character; (2) **Glyph** (Li et al., 2018; Eger et al., 2019a), which involves replacing characters with visually similar ones; (3) **Phonetic** (Le et al., 2022), which involves replacing characters in a way that makes the whole word sound similar to the original one. We present examples of these three perturbation rules in Table 1.

| Rule | Ori. Sentence | Adv. Sentence |
|---|---|---|
| Typo | A red ball on green grass under a blue sky. | A rde ball on green grass under a blue skky. |
| Glyph | A red ball on green grass under a blue sky. | A rêd ball 0n green grass under a blue sky. |
| Phonetic | A red ball on green grass under a blue sky. | A read ball on green grass under a blue SKY. |

Table 1: Examples of our perturbation rules.

### 3.3.2 GREEDY SEARCH

Given the efficiency and effectiveness of greedy algorithms in previous black-box text attack problems (Feng et al., 2018; Pruthi et al., 2019), we also employ a greedy algorithm here and organize it as the following steps.

**Step 1: word importance ranking.** Given a sentence of $n$ words $c = \{w_1, w_2, ..., w_n\}$, it is usually the case that only some keywords act as the influential factors for controlling DMs. Therefore, we aim to first identify such words and then perform attack. The identification of word importance is trivial in a white-box scenario, e.g., by inspecting model gradients (Behjati et al., 2019), but is challenging in the considered black-box setting. To address this, we directly measure the marginal influence of the word $w_i$ on the generation distribution via $I_{w_i} := \mathcal{D}(p_\theta(x|c\backslash w_i)\|p_\theta(x|c))$ where $c\backslash w_i = \{w_1, ..., w_{i-1}, w_{i+1}, ...w_n\}$ denotes the sentence without the word $w_i$ and $\mathcal{D}$ refers to the divergence measure defined earlier. With this, we can compute the influence score $I_{w_i}$ for each word $w_i$ in the sentence $c$, and then obtain a ranking over the words according to their importance.

**Step 2: word perturbation.** We then attempt to perturb the detected important words to find the adversarial example $c'$. Concretely, for the most important word $w_i \in c$, we randomly select one character in it and then randomly apply one of the meta-operations in the perturbation rule of concern, e.g., character swapping and deleting, to obtain a perturbed word as well as a perturbed sentence. Repeating this five times results in 5 perturbed sentences $\{c'_1, c'_2, ...c'_5\}$. We select the sentence leading to the highest generation divergence from the original sentence, i.e., $\mathcal{D}(p_\theta(x|c'_i)\|p_\theta(x|c)), \forall i \in \{1, ..., 5\}$ as the eventual adversarial sentence $c'$. If the attack has not reached the termination condition, the next word in the importance ranking will be selected for perturbation.

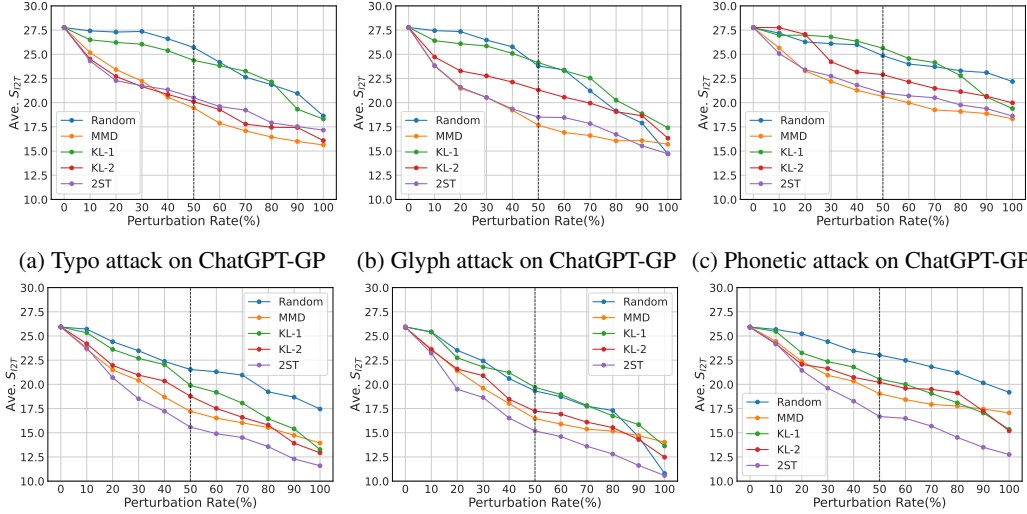

Figure 3: CLIP Score at different perturbation rates on ChatGPT-GP and SBU Corpus.

# 4 DIAGNOSTIC EXPERIMENTS

In this section, we provide diagnostic experiments consisting of two aspects: (1) assessing the four proposed attack objectives under varying perturbation rates; (2) analyzing which part of the DM is significantly misled. These analyses not only validate the efficacy of our method, but also deepen our understanding of the robustness of T2I DMs, and provide insightful perspectives for future works.

We consider two textual datasets – ChatGPT-GP and SBU corpus (Ordonez et al., 2011), as prompts for T2I generation. The victim model employed is Stable Diffusion (Rombach et al., 2022). Subsequently, we use $S_{T2I}$ and $S_{T2T}$ as evaluation metrics to separately compute the semantic disparity between text and image, as well as between different textual inputs. More details are described in Appendix B.1.

## 4.1 ATTACK WITH DIFFERENT OBJECTIVES

We first conduct a diagnostic experiment on the effects of the four proposed attack objectives under various perturbation rules. We define the perturbation rate as the ratio between the number of perturbed words and the total words in a sentence, and vary it from 0% to 100% with an interval of 10%. We calculate the average values of $S_{I2T}$ and $S_{T2T}$ on ChatGPT-GP and SBU Corpus, which are reported in Figure 3. Note that we also include a random baseline in comparison.

On ChatGPT-GP, all methods exhibit a declining trend in $S_{I2T}$ as the perturbation rate increases. Considering high perturbation rates rarely exist in practice, we primarily focus on situations where the perturbation rate is less than 50%. Within this range, we observe that the curves corresponding to MMD, KL-2, and 2ST display a rapid decrease across all three perturbation rules, more than 2× faster than random and KL-1 when using typo and glyph rules. It is also noteworthy that MMD and 2ST perform similarly and yield the best overall results.

On SBU Corpus, it is evident that 2ST is more effective than MMD. Additionally, even with a perturbation rate of 100%, the random method fails to achieve a similar $S_{I2T}$ score compared to other methods. This observation suggests the effectiveness of our 2-step attack algorithm. Additionally, glyph-based perturbations lead to the most rapid decrease in performance, followed by typo perturbations, and phonetic perturbations lead to the slowest drop. This disparity may be attributed to glyph perturbations completely disrupting the original word embedding.

## 4.2 WHICH PART OF THE DM IS SIGNIFICANTLY MISLED?

Previous studies suggest that attacking only the CLIP encoder is sufficient for misleading diffusion models (Zhuang et al., 2023). However, our method is designed to attack the entire generation process instead of the CLIP encoder. For empirical evaluation, we conduct a set of experiments in this section.

We include two additional attack methods: attacking only the CLIP encoder and attacking only the diffusion process. Regarding this first one, we focus solely on maximizing the dissimilarity between

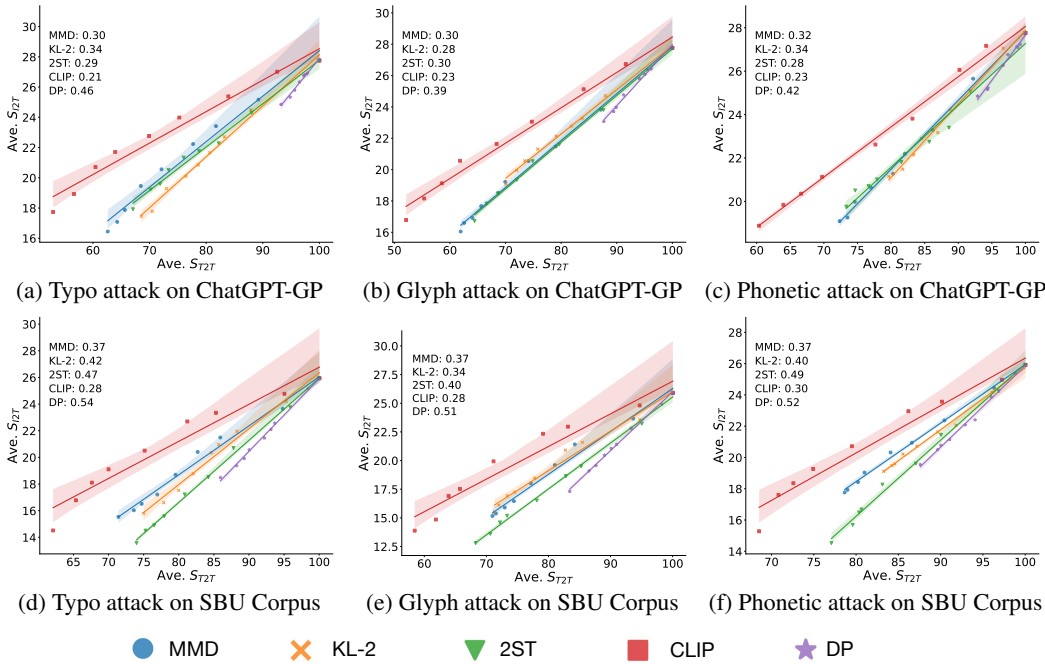

Figure 4: Corelation between $S_{I2T}$ and $S_{T2T}$ on ChatGPT-GP and SBU Corpus. The numbers in the upper-left corner represent the slopes of the plotted lines.

the original text and the adversarial one. To achieve this, we employ $S_{T2T}$ as the optimization objective, i.e., $\mathcal{D}_{\text{CLIP}} = S_{T2T} = \max(0, 100 \cdot g_\phi(c)^\top g_\phi(c'))$. As for the second one, we modify Eq. (4) and devise a new attack objective as follows ($\alpha$ and $\beta$ denote two trade-off coefficients):

$$\mathcal{D}_{\text{DP}} \approx \left[\alpha g_\phi(c') - \beta \frac{1}{N} \sum_{i=1}^{N} h_\phi(x_i)\right]^\top g_\phi(c). \tag{8}$$

While maximizing the distance between the original text and the adversarial images, we also aim to ensure that the representations of the adversarial text and the original text are as similar as possible. This confines that even though the entire DM is under attack, the CLIP encoder remains safe. More details of this equation can be found in Appendix A.2.

Given the poor performance of the random and KL-1 methods, we exclude them from this study. Considering that high perturbation rates are almost impossible in the real world, we experiment with perturbation rates only from 0% to 80%. We compute the average $S_{I2T}$ and $S_{T2T}$ across all texts at every perturbation rate, and plot their correlations in Figure 4.

As shown, exclusively targeting the CLIP encoder during the attack process yields the maximum slope of the regression line, while solely attacking the diffusion process leads to the minimum slope. For instance, in the typo attack on ChatGPT-GP, the attack method solely attacking the CLIP encoder exhibits the lowest slope of 0.21, whereas the attack method exclusively targeting the diffusion process shows the highest slope of 0.46. Attack methods that simultaneously target both processes display slopes between these extremes. These clearly support that our attack objectives simultaneously attack the CLIP encoder and the diffusion process. Furthermore, through the slope information, we can conclude that directly attacking the diffusion process yields a more significant decrease in image-text similarity at a given textual semantic divergence. Across all datasets and perturbation spaces, the slope disparity between direct attacks on the diffusion process and direct attacks on the CLIP encoder is mostly above 0.1, and the maximum slope disparity reaches even 0.15.

### 4.3 COMPARE WITH NON-DISTRIBUTION ATTACK OBJECTIVE

We conduct a comparison experiment between our distribution-based optimization objective, referred to as **2ST**, and a non-distribution method that solely relies on the $S_{T2I}$ of the prompt combined with a single definite image (DI). Following Section 4, we randomly sampled 20 texts from ChatGPT-GP and SBU Corpus separately, then applied typo rule to perturb sampled texts with different perturbation rates. The results, depicted in Figure 5, clearly demonstrate the superior effectiveness of the distribution-based approach.

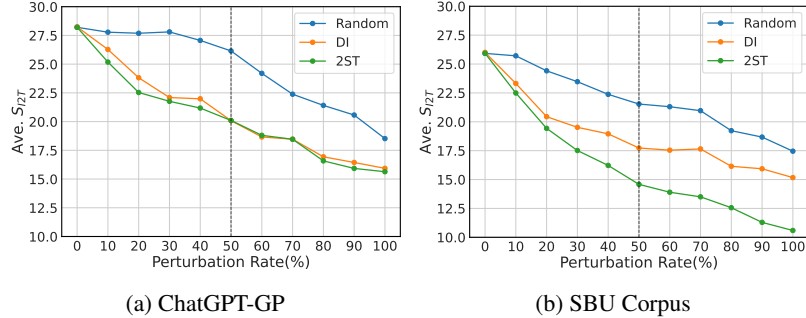

|                | (a) ChatGPT-GP | | (b) SBU Corpus |

Figure 5: CLIP Score at different perturbation rates on ChatGPT-GP and SBU Corpus with typo rule.

# 5 REAL-WORLD ATTACK EXPERIMENT

Based on the preceding analysis, we identify that 2ST and MMD are two good attack objectives for T2I DMs. In this section, we will carry out attacks in real-world scenarios, where termination conditions are incorporated to balance the perturbation level and effectiveness.

For a more comprehensive evaluation of our attack methodology under realistic scenarios, we integrate two additional datasets, namely DiffusionDB (Wang et al., 2022b), and LAION-COCO (Schuhmann et al., 2021). We employ four evaluation metrics: *L-distance* quantifying the extent of text modifications, *Ori./Adv.* $S_{I2T}$ gauging the similarity between the original text and original/adversarial images, *average query times* serving as an efficiency metric for the attack, and *human evaluation* appraising the inconsistency between the image and text. More details can be found in Appendix B.2.

| Dataset | Attacker | Ori. $S_{I2T}$ | Ave. Len. | L-distance | Adv. $S_{I2T}$ | Ave. Query | Hum. Eval. |
|---------|----------|----------------|-----------|------------|----------------|------------|------------|
| ChatGPT-GP | Typo | 27.61±2.07 | 10.41 | 2.92 | 23.21±3.08 | 19.43 | 84.34% |
| | Glyph | | | 2.27 | 23.09±2.75 | 18.63 | 84.65% |
| | Phonetic | | | 5.38 | 22.67±3.58 | 17.78 | 86.16% |
| DiffusionDB | Typo | 29.17±3.36 | 10.62 | 2.29 | 22.70±3.31 | 17.25 | 76.64% |
| | Glyph | | | 1.81 | 22.71±3.22 | 16.30 | 76.64% |
| | Phonetic | | | 5.04 | 22.91±3.34 | 16.27 | 75.51% |
| LAION-COCO | Typo | 27.54±2.86 | 9.17 | 2.08 | 21.73±3.62 | 14.77 | 80.21% |
| | Glyph | | | 1.85 | 21.32±3.69 | 15.11 | 81.89% |
| | Phonetic | | | 5.04 | 21.76±3.87 | 16.15 | 79.32% |
| SBU Corpus | Typo | 24.99±3.43 | 11.69 | 2.97 | 19.65±3.53 | 21.19 | 84.34% |
| | Glyph | | | 2.42 | 19.01±3.76 | 20.54 | 85.41% |
| | Phonetic | | | 5.85 | 18.86±3.91 | 19.92 | 85.41% |

Table 2: Real-world attack with the **2ST** attack objective.

| Dataset | Attacker | Ori. $S_{I2T}$ | Ave. Len. | L-distance | Adv. $S_{I2T}$ | Ave. Query | Hum. Eval. |
|---------|----------|----------------|-----------|------------|----------------|------------|------------|
| ChatGPT-GP | Typo | 27.61±2.07 | 10.41 | 1.77 | 24.54±2.69 | 14.17 | 84.21% |
| | Glyph | | | 1.15 | 24.88±2.67 | 13.08 | 84.36% |
| | Phonetic | | | 3.81 | 26.08±2.21 | 14.58 | 80.02% |
| DiffusionDB | Typo | 29.17±3.36 | 10.62 | 1.75 | 24.94±3.82 | 13.72 | 72.77% |
| | Glyph | | | 1.29 | 24.81±3.90 | 13.41 | 73.53% |
| | Phonetic | | | 4.27 | 26.71±3.24 | 15.13 | 70.09% |
| LAION-COCO | Typo | 27.54±2.86 | 9.17 | 1.75 | 23.04±4.10 | 13.33 | 80.21% |
| | Glyph | | | 1.35 | 23.72±3.91 | 12.35 | 82.04% |
| | Phonetic | | | 3.62 | 25.06±3.09 | 13.21 | 77.37% |
| SBU Corpus | Typo | 24.99±3.43 | 11.69 | 1.91 | 21.37±3.92 | 16.36 | 82.05% |
| | Glyph | | | 1.37 | 21.44±3.66 | 15.01 | 82.33% |
| | Phonetic | | | 3.72 | 23.15±3.25 | 16.20 | 79.67% |

Table 3: Real-world attack with **MMD distance** attack objective.

We first conduct the real-world attack experiment on Stable Diffsuion. Table 2 and Table 3 present the results of our real attack experiments using various perturbation rules on different datasets, with 2ST and MMD distance as the attack objectives, respectively. Since the termination criteria for the two optimization algorithms differ, we cannot compare them directly. Considering that our method

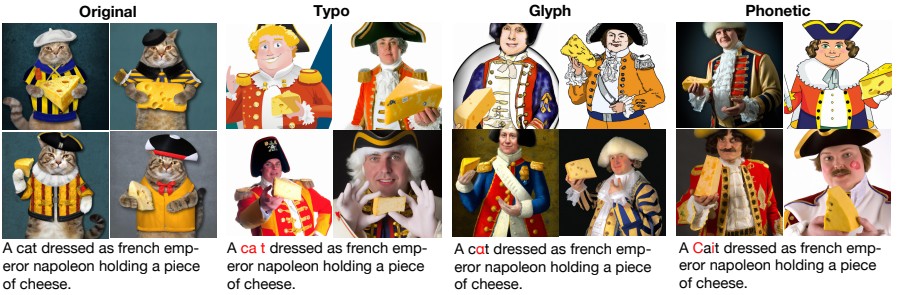

Figure 6: An illustration of adversarial attack against DALL-E 2 with **MMD** attack objective.

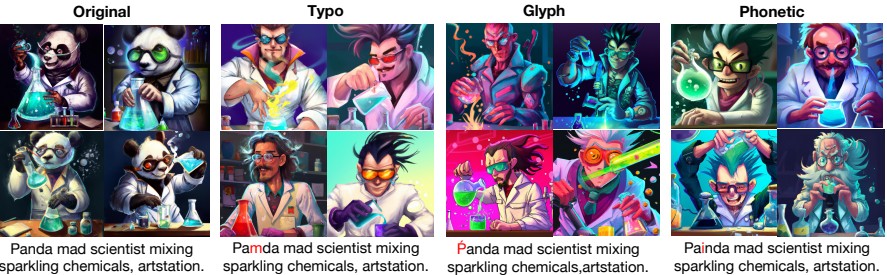

Figure 7: An illustration of adversarial attack against DALL-E 2 with **2ST** attack objective.

involves querying each word of the sentence (described in Section 3.3.2), the query times minus the sentence length, which we named *true query times*, can better demonstrate the true efficiency of our approach. From this perspective, our method requires less than 10 *true query times* to achieve more than 4 $S_{I2T}$ score drop across most datasets. The *human evaluation* score is no less than 75%. Simultaneously, we observe that our modifications are relatively minor. In the typo and glyph attacker, we require an *L-distance* of less than 3, while in the phonetic attacker, the threshold remains below 6. Furthermore, ChatGPT-GP and LAION-COCO are more susceptible to our attack, possibly attributed to their clearer sentence descriptions and improved flow in the text. In conclusion, with minimal modifications and a limited number of queries to the model, we achieve a significant decrease in text-image similarity, substantiated by human evaluations. We also conducted an experiment for single-image guidance attack in the real-attack experiment in Table 5 in Appendix C.3.

**DALL-E 2** (Ramesh et al., 2022) is also a powerful image generation model that can create realistic and diverse images from textual descriptions. We then conduct a case study with the same attack method used in Stable Diffusion. The results respectively obtained with the attack objective MMD and 2ST are presented in Figure 6 and Figure 7. More cases can be found in Appendix C.1.

Furthermore, we carry out experiments on the **defense sides** in Appendix C.2. Table 4 shows that auto-typo correctors can partially correct human mistakes from typo perturbations in our work. However, correcting a little fiercely perturbed sentence caused by glyphs and phonetics proves challenging. Hence, our method remains effective against auto-typo correctors. It is also worth noting that these correctors often struggle to automatically rectify words in a manner that aligns with the user's intent when human intervention is not involved in the word selection process from the correction word list.

Finally, we engage in a discussion concerning **human attack** without algorithmic interventions and **word-level attacks** in Appendix D. We discuss the challenges encountered in attacking without algorithmic interventions, providing evidence for the effectiveness of our attack algorithm. Additionally, we explain why this study did not employ word-level attacks and highlight the impracticality of directly transferring text-based adversarial attacks to DMs.

## 6 CONCLUSION

In this work, we present a comprehensive evaluation of the robustness of DMs against real-world attacks. Unlike previous studies that focused on malicious alterations to input texts, we explore an attack method based on realistic errors that humans can make to ensure semantic consistency. Our novel distribution-based attack method can effectively mislead DMs in a black-box setting without any knowledge of the original generative model. Importantly, we show that our method does not solely target the text encoder in DMs, it can also attack the diffusion process. Even with extremely low perturbation rates and query times, our method can achieve a high attack success rate.

## ETHICS STATEMENT

A potential negative societal impact of our approach is that malicious attackers could exploit it to construct targeted attacks by modifying the loss function, leading to the generation of unhealthy or harmful images, thus causing security concerns. As more people focus on T2I DMs due to their excellent performance on image generation. In such scenarios, it becomes inevitable to address the vulnerability of DMs which can be easily attacked through black-box perturbation. Our work emphasizes the importance for developers of DMs to consider potential attacks that may exist in real-world settings during the training process.

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

# A  PROOF OF EQUATIONS

## A.1  PROOF OF EQ. (3)

$$
\mathcal{D}_{\mathrm{KL}}(p_\theta(x|c')\|p_\theta(x|c)) \approx \mathcal{D}_{\mathrm{KL}}(p_\theta(x|c')\|p_\phi(x|c))
$$

$$
= \mathbb{E}_{p_\theta(x|c')}\left[\log\frac{p_\theta(x|c')}{p_\phi(x|c)p(c)} + \log p(c)\right] \tag{9}
$$

$$
= \mathbb{E}_{p_\theta(x|c')}[-e_\phi(x,c)] + \mathbb{E}_{p_\theta(x|c')}[\log p_\theta(x|c')] + C
$$

## A.2  PROOF OF EQ. (8)

We demonstrate in this section why Eq. (8) represents only the attack diffusion process. For Eq. (8), we can expand it as:

$$
\mathcal{D}_{\mathrm{DP}} \approx \left[\alpha g_\phi(c') - \beta\frac{1}{N}\sum_{i=1}^{N}h_\phi(x_i)\right]^\top g_\phi(c)
$$

$$
= \alpha g_\phi(c')^\top g_\phi(c) - \beta\left[\frac{1}{N}\sum_{i=1}^{N}h_\phi(x_i)\right]^\top g_\phi(c) \tag{10}
$$

where $c$ is the original text, $c'$ is the modified text with $x_i$ generated from it. The first term, $\alpha g_\phi(c')^\top g_\phi(c)$, measures the similarity between the original text and the adversarial text. The second term, $\beta\left[\frac{1}{N}\sum_{i=1}^{N}h_\phi(x_i)\right]^\top g_\phi(c)$, represents the similarity between the original text and the adversarially generated images. Maximizing this objective constrains the original text and the adversarial text to be as similar as possible after being encoded by the text encoder, while minimizing the similarity between the original text and the adversarial generated images. In this way, it avoids attacking the text encoder and solely attacks the diffusion process.

Since Eq. (8) is modified from Eq. (4), we also provide an expanded explanation for Eq. (4) as follows:

$$
\mathcal{D}_{\mathrm{KL}}(p_\theta(x|c')\|p_\phi(x|c)) \approx \alpha\left[\frac{1}{N}\sum_{i=1}^{N}h_\phi(x_i)\right]^\top\left(g_\phi(c') - g_\phi(c)\right)
$$

$$
= \alpha\left[\frac{1}{N}\sum_{i=1}^{N}h_\phi(x_i)\right]^\top g_\phi(c') - \alpha\left[\frac{1}{N}\sum_{i=1}^{N}h_\phi(x_i)\right]^\top g_\phi(c) \tag{11}
$$

The first term, $\alpha\left[\frac{1}{N}\sum_{i=1}^{N}h_\phi(x_i)\right]^\top g_\phi(c')$, measures the similarity between the adversarial text and adversarially generated images. The second term, $\alpha\left[\frac{1}{N}\sum_{i=1}^{N}h_\phi(x_i)\right]^\top g_\phi(c)$, represents the similarity between the original text and the adversarial generated images. Maximizing this objective constrains the encoded adversarial text and the adversarial images to be as similar as possible, which essentially ensures the quality of the text embedding guided image diffusion process, while minimizing the similarity between the original text and the adversarial generated images. In this way, it avoids attacking the diffusion process and solely attacks the text encoder, different from Eq. (8).

# B  EXPERIMENT DETAIL

## B.1  DIAGNOSTIC EXPERIMENTS

**Datasets.** We consider two types of textual data for prompting the generation of T2I DMs: (1) 50 ChatGPT generated (ChatGPT-GP) prompts by querying: "generate 50 basic prompts used for image synthesis." and (2) 50 image captions from SBU Corpus (Ordonez et al., 2011). Such a dataset facilitates a thorough investigation of the efficacy and applicability of our method in practical image-text generation tasks.

**Victim Models.** We choose Stable Diffusion (Rombach et al., 2022) as the victim model due to its widespread usage, availability as an open-source model, and strong generation capability. Stable Diffusion utilizes a denoising mechanism that operates in the latent space of images and incorporates cross-attention to leverage guidance information. Text inputs are first processed by CLIP's text encoder to generate text embeddings, which are subsequently fed into the cross-attention layers to aid in image generation.

**Evaluation Metrics.** We use the CLIP Score (Hessel et al., 2021), esentially the aforementioned $h_\phi(x)^\top g_\phi(c)$, to measure the semantic similarity between the original text $c$ and the generated images $\{x_1, \ldots, x_N\}$ based on the adversarial text $c'$. Specifically, we define the metric $S_{I2T} = \frac{1}{N} \sum_{i=1}^{N} \max(0, 100 \cdot g_\phi(c)^\top h_\phi(x'))$ over the generated images, and we hypothesize that a higher $S_{I2T}$ indicates a less adversarial text $c'$. Typically, $N$ is set to 15 to balance efficiency and fidelity. We can also calculate the similarity between the original text $c$ and the adversarial text $c'$ with $S_{T2T} = \max(0, 100 \cdot g_\phi(c)^\top g_\phi(c'))$. Though these two metrics use the same notations as our attack objectives, we actually use various pre-trained CLIP to instantiate them to avoid over-fitting. In particular, we employ the CLIP with VIT-L-patch14 backbone for attack while using VIT-L-patch14-336 for evaluation.

### B.2 REAL-WORLD ATTACK EXPERIMENT

**Datasets.** To provide a more comprehensive evaluation of our attack method in realistic scenarios, we incorporate two additional datasets. The first one, DiffusionDB (Wang et al., 2022b), is a large-scale dataset of 14 million T2I prompts. The second one, LAION-COCO (Schuhmann et al., 2021), includes captions for 600 million images from the English subset of LAION-5B (Schuhmann et al.). The captions are generated using an ensemble of BLIP L/14 (Li et al., 2022) and two CLIP (Radford et al., 2021) variants. To conjoin diversity and efficiency, we randomly select 100 examples from each of the aforementioned datasets. Additionally, we also increased the size of ChatGPT-GP and SBU to 100 for this experiment.

**Attack method.** As said, we consider attacking based on MMD and 2ST. A threshold on the value of $\mathcal{D}$ is set for termination. If it is not reached, the attack terminates at a pre-fixed number of steps.

**Evaluation metric.** We use four metrics to evaluate our method in real-world attack scenes. (1) Levenshtein distance (L-distance), which measures the minimum number of single-character edits, a powerful indicator of the number of modifications made to a text. (2) Ori.$S_{I2T}$ and Adv.$S_{I2T}$ which indicate the similarity between the original text and original images as well as that between the original text and the adversarial images respectively. The mean and variance are both reported. (3) Average query times, which represents the number of times that DM generates images with one text, and serves as a metric for evaluating the attack efficiency. (4) Human evaluation, where humans are employed to assess the consistency between the image and text. Let $N_1$ represent the number of images generated by the original text that are consistent with the original text, and $N_2$ represent the number of images generated by the adversarial text that are consistent with original text. If $(N_2 - N_1) > 1$, the attack on that particular prompt text is deemed meaningless. Let's assume the frequency of samples where $(N_2 - N_1) > 1$ as $N_u$, and the effective total number of samples should be $N_{\text{total}} - N_u$. If $(N_1 - N_2 > 1)$, it indicates a successful attack. We use $N_c$ to represent the number of samples where the attack is successful. Thus, the final score for each evaluator is given by $N_c/(N_{\text{total}} - N_u)$. The average of three human annotators represents the overall human evaluation score (Hum.Eval).

## C EXPERIMENT RESULT

### C.1 CASE STUDY ON DALL-E 2

As a supplement to the case study experiments on DALL-E 2 in Section 5, we present two additional cases for each of the optimization objectives, MMD and 2ST, shown in Figure 8.

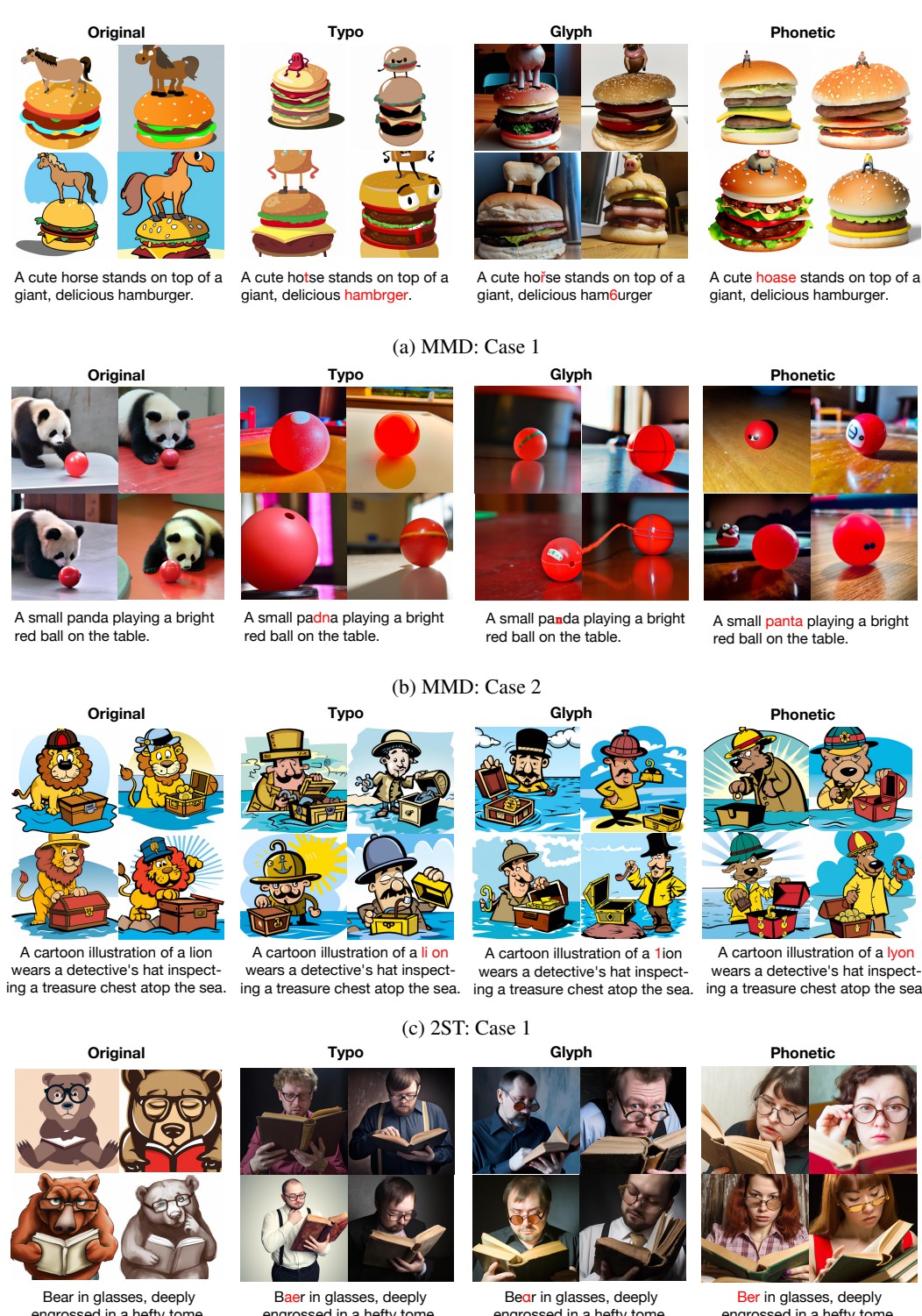

Figure 8: Illustrations of adversarial attack against DALL-E 2 with MMD or 2ST attack objective. Each of these objectives has two cases.

## C.2 Evaluation on the Defense Side

We used two widely used correctors, LanguageTool[1] and Online Correction[2] as the corrector. Then we selected 100 successfully attacked text samples for each perturbation rule with 2ST attack objective from the ChatGPT-GP dataset, following the settings outlined in Section B.1. Finally, we evaluated the samples modified by these typo-correctors to determine whether they were successfully repaired. Note that if the corrector provided more than one recommended correct word, we utilized the first recommended word. Table 4 presents the comparison on **Successful Repair Rate (SRR)**.

| Typo Corrector | Perturbation Rule | SRR |
|---|---|---|
| LanguageTool | Typo | 68% |
| | Glyph | 39% |
| | Phonetic | 21% |
| Online Correction | Typo | 81% |
| | Glyph | 42% |
| | Phonetic | 25% |

Table 4: SRR of auto-correctors to 3 perturbation rules.

Large language models are indeed effective in mitigating character-level attacks. We employed ChatGPT to repair samples generated by three attack methods: typo, glyph, and phonetic. With 50 samples for each method, the SRR reached 91.3%. However, due to the high cost, large models are less suitable for correcting the input for image generation models by ordinary users.

## C.3 Real-word Attack With DI Attack Objective

| Dataset | Attacker | Ori. $S_{I2T}$ | Ave. Len. | L-distance | Adv. $S_{I2T}$ | Ave. Query | Hum. Eval. |
|---|---|---|---|---|---|---|---|
| ChatGPT-GP | Typo | 27.61±2.07 | 10.41 | 2.65 | 24.61±3.32 | 16.97 | 79.36% |
| | Glyph | | | 2.09 | 24.94±3.01 | 16.86 | 77.45% |
| | Phonetic | | | 5.13 | 26.67±3.74 | 15.71 | 78.68% |

Table 5: Real-world attack with the **DI** attack objective.

We conduct a real-world attack with the DI attack objective(i.e. single-image guidance attack) on the ChatGPT-GP dataset, employing the same settings as the other objectives detailed in Section 5. Table 5 illustrates that the $Adv.S_{I2T}$ and human evaluation scores associated with this objective are relatively low. This observation suggests that the attack with the DI objective may be susceptible to overfitting on a single image.

# D Discussion on Human Attack and Word-level Attack

## D.1 Human Attack

Firstly, we would like to emphasize the effectiveness of our adversarial optimization algorithm. In order to demonstrate this, we compare our method with human attack without algorithmic interventions. We randomly selected a set of sentences and made random modifications to the most important words based on human intuition. Remarkably, we observed that a lot of sentences with these modifications did not result in DMs generating incorrect images. This further substantiates the effectiveness of our attack algorithm. We present two illustrative cases in Figure 9. The results emphasize the difficulty of this attack task and show the effectiveness of our method.

## D.2 Word-level Attack

Then we talk about the other level attacks such as word-level attack. Due to the high sensitivity of the DM to individual words in the prompt, word-level attacks such as synonym replacement or context

---

[1]https://languagetool.org/

[2]https://www.onlinecorrection.com/

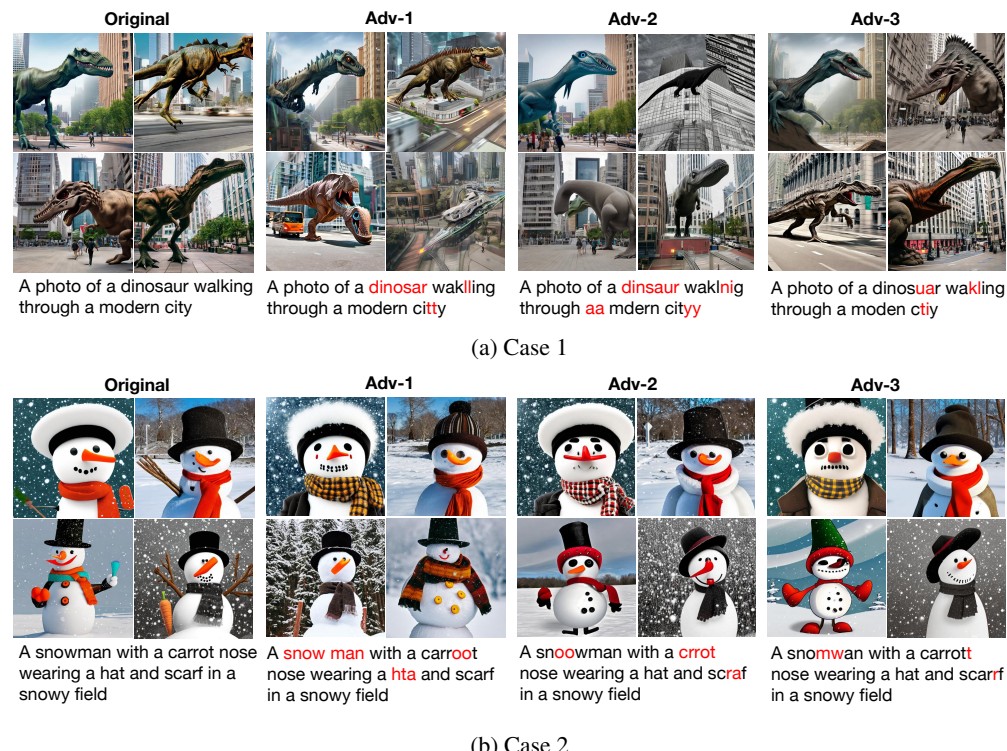

(a) Case 1

(b) Case 2

Figure 9: Illustrations of human attack method against Stable Diffusion. Adversarially modified content is highlighted in red.

filling were not employed in this study. If we were to use synonym replacements and substitute words with less commonly used ones, those words themselves might have multiple meanings. In such cases, the model is likely to generate images based on alternative meanings, making the substituted words different in the context of the sentence, even though they may be synonyms in terms of individual words. Therefore, a more stringent restriction is required for word-level replacements. It is precisely because of this reason that traditional text-based attack methods are not applicable to image-text generation. For instance, in sentiment classification tasks, they only consider the overall sentiment of the entire sentence, and the ambiguity of a particular word does not significantly impact the overall result. However, this sensitivity becomes crucial in the context of T2I. Hence, further research is needed to explore word-level and sentence-level attacks on T2I generation models.

| Attack Method | Perturbation Level | DCR |
|---|---|---|
| BERTAttack | Word-level | 19% |
| PWWS | Word-level | 24% |
| Our Method(Typo) | Char-level | 82% |
| Our Method(Glyph) | Char-level | 96% |
| Our Method(Phonetic) | Char-level | 73% |

Table 6: Comparison on DCR between word-level attacks and our method.

To better illustrate our point, we conducted a comparison on the semantic consistency between word-level attack and our method. We chose two classical textual word-level adversarial attack algorithms in natural language processing(NLP), BERTAttack (Li et al., 2020) and PWWS (Ren et al., 2019) to compare with our method with 3 perturbation rules (typo, glyph and phonetic) with same optimization objective. We sampled 100 texts from successfully attacked texts for each attack method and evaluated the description consistency between these adversarial texts and their corresponding original texts by humans. To avoid bias, we evaluated each text with three people and took the plural of the three people's opinions as the final decision. Table 6 below presents the comparison on **Description Consistency Rate (DCR)** and shows that our method based on character level

perturbation can keep the description consistency far more than word-level attack methods such as BERTAttack and PWWS.

We also list some examples generated by word-level adversarial attack methods in table 7. It is evident that significant semantic changes have occurred in the examples presented. Therefore, word-level attacks in still have a long way to go in T2I adversarial attack.

| Attack Method | Ori. Text | Adv. Text |
|---|---|---|
| | A red ball on green grass under a blue sky. | A red field on green grass under a blue sky. |
| BERTAttack | A white cat sleeping on a windowsill with a flower pot nearby. | A green cat sleeping on a windowsill with a flower pot nearby. |
| | A wooden chair sitting in the sand at a beach. | A wooden camera sitting in the sand at a beach. |
| | A red ball on green grass under a blue sky. | A red orchis on green grass under a blue sky |
| PWWS | A white cat sleeping on a windowsill with a flower pot nearby. | A white guy sleeping on a windowsill with a flower pot nearby |
| | A wooden chair sitting in the sand at a beach. | A wooden chairwoman sitting in the baroness at a beach. |

Table 7: Word-level attack examples by BERTAttack and PWWS with **2ST** attack objective.

### D.3 HUMAN EVALUATION USING ADVERSARIAL TEXT

To prove that human evaluation won't be affected by the adversarial noise of text. We conduct the experiment to evaluate the difference between the image content and the actual meaning of the adversarial text. We let $N_1$ represent the number of images generated by the original text that are consistent with the original text as original, while $N_2$ represents the number of images generated by the adversarial text that are consistent with the **adversarial text**. The new overall human evaluation score is calculated in the same way as the original paper. Table 8 shows the new overall human evaluation on the same adversarial dataset generated under real-world attack with the 2ST attack objective in section 5. We found that although the new overall human evaluation score will decrease to some extent compared to the old score, the change is not significant, indicating that human evaluations are not affected by the adversarial text.

| Dataset | Attacker | Ori. $S_{I2T}$ | Ave. Len. | L-distance | Old Hum. Eval. | New Hum. Eval. |
|---|---|---|---|---|---|---|
| | Typo | | | 2.92 | 84.34% | 82.19% |
| ChatGPT-GP | Glyph | 27.61±2.07 | 10.41 | 2.27 | 84.65% | 83.42% |
| | Phonetic | | | 5.38 | 86.16% | 82.06% |
| | Typo | | | 2.29 | 76.64% | 74.34% |
| DiffusionDB | Glyph | 29.17±3.36 | 10.62 | 1.81 | 76.64% | 75.26% |
| | Phonetic | | | 5.04 | 75.51% | 73.61% |
| | Typo | | | 2.08 | 80.21% | 78.28% |
| LAION-COCO | Glyph | 27.54±2.86 | 9.17 | 1.85 | 81.89% | 80.94% |
| | Phonetic | | | 5.04 | 79.32% | 77.02% |
| | Typo | | | 2.97 | 84.34% | 82.13% |
| SBU Corpus | Glyph | 24.99±3.43 | 11.69 | 2.42 | 85.41% | 85.57% |
| | Phonetic | | | 5.85 | 85.41% | 81.95% |

Table 8: Real-world attack with the **2ST** attack objective.

### D.4 LEXICAL PROPERTIES OF THE MODIFIED WORD

The modified words of adversarial text can be nouns, adjectives and verbs. In descriptive text about objects, there is a greater occurrence of modified words in nouns. It is important to note that our approach aims to identify words that are most important for the model, rather than those deemed most important by humans. Therefore, in theory, it is not limited to a specific part of speech. Figure 10 shows some cases without noun modification.

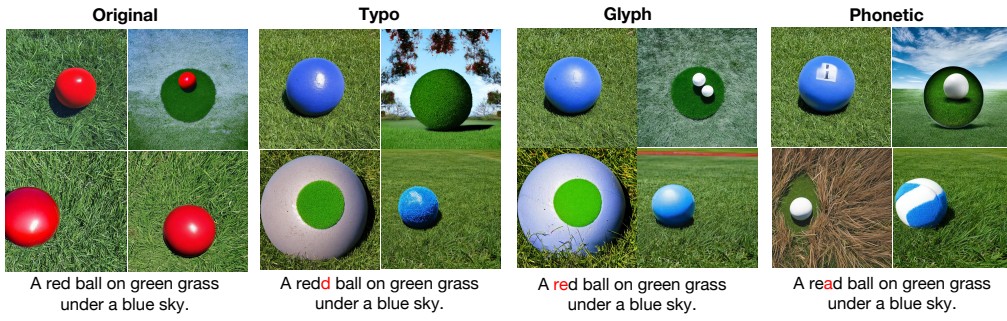

(a) Case with modified adjectives.

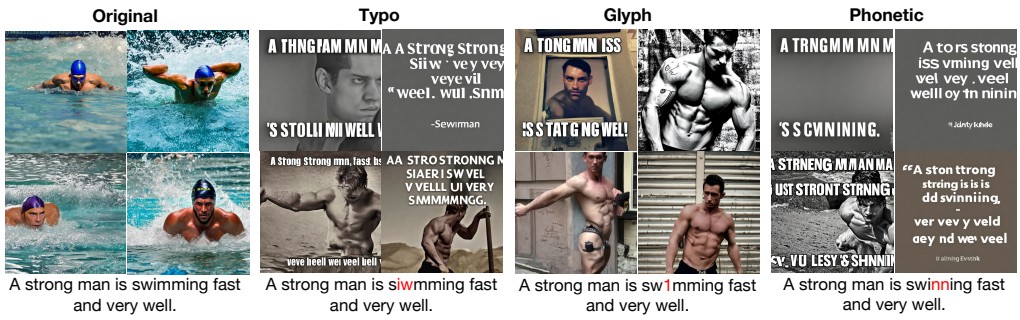

(b) Case with modified verbs.

Figure 10: Cases without noun modification.

## E  LIMITATION

In our experiments, we employ DMs as the testbed and evaluate both random attack methods and our proposed method with four optimization objectives on our custom benchmark datasets. Due to limited resources, we focus on Stable Diffusion for the complete experiment and DALL-E 2 for the case study, given that our method involves 12 combinations. Therefore, conducting more comprehensive experiments covering different model architectures and training paradigms is a direction for future research.

## F  COMPUTE DEVICE

All experiments were conducted on NVIDIA Tesla A100 GPUs. For diagnostic experiments, each attack rule with each optimization objective on one dataset took approximately 4 GPU days. For real-world attack experiments, each attack rule with each optimization objective on one dataset took approximately 3 GPU days. So in total, running all of the experiments (including ablation studies and case studies) requires about 250 GPU days.

