# OpenReview forum: "Evaluating the Robustness of Text-to-image Diffusion Models against Real-world Attacks"
_ICLR.cc/2024/Conference — Submitted to ICLR 2024_

### Official Review · Reviewer_BYdd · 2023-10-22

**Soundness:** 4 excellent
**Presentation:** 4 excellent
**Contribution:** 3 good
**Rating:** 8
**Confidence:** 3

**Summary:**

The paper considers the scenario of attacking text-to-image diffusion models with realistic modifications of the input text. Authors perform comprehensive analysis of possible attack objectives, provide thorough theoretical justification and evaluate them with extensive experiments. The paper argues that popular text-to-image models such as Stable Diffusion are vulnerable to this type of attacks thus pointing out the importance of further studying this topic.

**Strengths:**

1. Originality: the authors study a novel problem setting. I am not aware of other works providing comprehensive studies of considered attack objectives for T2I models.

2. Quality: conclusions in the paper are based on solid experimental justification, every step in excluding some of the objectives from further experiments is clearly justified. Experimental evaluation is rich and considers different metrics. Besides, the authors provide code for their experiments (although I did not examine it carefully). Results for Stable Diffusion model are provided which is relevant for the community due to its popularity.

3. Clarity: the paper is easy to follow, numerous illustrations facilitate understanding.

4. Significance: T2I approaches have significant impact in both research and industry, thus studying their possible vulnerabilities is valuable for the community.

**Weaknesses:**

1. Looking at the results in Figures 1, 6 and 7 one can observe that by analysing the output image one can see which word contains an error: “ice” or “cream” in Figure 1, “cat” in Figure 6 or “panda” in Figure 7. This clearly follows from the attack design selecting significant words (Section 3.3.2). However, this limits the “imperceptability” of the attacks. Modifying not the most significant words and still being able to perturb the image would be another interesting track.

**Questions:**

1. The results provided in the paper carry theoretical significance for understanding the work and limits of text-to-image models. This outcome is valuable on its own. However, it is not clear whether proposed attacks pose non-negligible security threat for some real-world applications. Up to my understanding, work with T2I models is usually not automated and controlled by a human being at all times: from providing the text input to controlling the output image. Thus a person seeing not what they were expecting to see will probably just inspect the textual query and modify it to delete any unwanted modifications. Are there any scenarios where a malicious attacker can really cause any harm with these attacks? An explicit analysis of this issue would provide additional context for the results and broader impact of this work.

---

> ### Author Response · Authors · 2023-11-17
> **Response to Reviewer BYdd**
>
> Thank you for the supportive comments. Below we address the detailed comments and hope that you may find our response satisfactory.
>
> **Question 1: The method limits the “imperceptability” of the attacks.**
>
> Thank you for kindly pointing it out. We make the following clarification in two perspectives. On one hand, for users, the mistakes made by themselves are imperceptible for the users because they did not notice these mistakes. Some kinds of perturbations (e.g., glyph and phonetic) are more imperceptible but can lead to significantly different images, as shown in Figures 1, 6, and 7. On the other hand, since the diffusion models typically rely on context to generate images, minor perturbations usually do not impact the model's understanding of the semantic meaning of the sentence. As demonstrated in Figure 9 in the Appendix, the diffusion model can still generate realistic and aligned images despite perturbing many words in the text prompt. Therefore, for smaller perturbations like those in our case, diffusion models are expected to produce images that resemble the original ones. However, our perturbed prompts successfully induced diffusion models to generate mismatched images.
>
> Besides, we generally adopt the perturbation space introduced in the field of NLP attacks. These perturbations are deemed more imperceptible for text than other kinds of perturbations (word-level, sentence-level). Nevertheless, as our proposed method is generally applicable to any perturbation space, we could further extend it to generate more imperceptible perturbations if a more appropriate perturbation space is developed.
>
> **Question 2: Modifying not the most significant words and still being able to perturb the image would be another interesting track.**
>
> Thank you for your thoughtful suggestion. We have indeed taken this into consideration. The important words identified by our method may not universally align with what is considered significant to humans. Sometimes our method also selects words that are not significant to human understanding. We also conduct the human attack experiment in Appendix D, manually selecting words considered to be of high importance by the human. However, the actual outcomes were not entirely satisfactory. We will consider your suggestion and explore methods for perturbing sentences while adhering to constraints on words deemed important by human evaluation in future work.
>
> **Question 3: Are there any scenarios where a malicious attacker can really cause any harm with these attacks. An explicit analysis of this issue would provide additional context for the results and broader impact of this work.**
>
> Thank you for your kind advice. As described in the introduction, our proposed attack aims to evaluate the robustness of text-to-image diffusion models under real-world perturbations rather than generating harmful images. Therefore, the attacker is more likely to be the model developer rather than a malicious attacker. Evaluating the robustness of diffusion models in our setting is also crucial for real-world deployment, which can not only improve user efficiency but also cut down on user costs by eliminating the need to revisit and rectify mistakes in prompts, especially when dealing with bulk operations. We will further extend our method to generate harmful content in future work.

---

> > ### Comment · Reviewer_BYdd · 2023-11-21
> > **Reply**
> >
> > Thank you for your reply. The significance of the obtained results is still not convincing for me. I did not mean generating harmful images. I meant situations where significant harm could be caused by a user falling into one of the proposed attack scenarios. I don’t think T2I applications are safety-critical like, for example, autonomous driving. Therefore, T2I models being not robust does not seem like a critical issue.
> >
> > “Evaluating the robustness of diffusion models in our setting is also crucial for real-world deployment, which can not only improve user efficiency but also cut down on user costs by eliminating the need to revisit and rectify mistakes in prompts, especially when dealing with bulk operations.”
> >
> > I do not really see how evaluating robustness can help improve user efficiency. Showing that T2I models are not robust suggests that some sort of preprocessig for the prompt has to be used like spell-checking, glyph-checking etc. Such results allow to raise awareness but do not really improve the efficiency.

---

> ### Author Response · Authors · 2023-11-22
>
> Thank you for your follow-up comment.
> Below we address the detailed concerns. We hope that you may find our response satisfactory.
>
> **Question 1:  T2I models being not robust does not seem like a critical issue.**
>
> Thanks for the comments and we make the following clarifications. As described in the introduction, our paper aims to evaluate the robustness of text-to-image diffusion models in the face of real-world perturbations. Such perturbations may arise from human errors and could lead to the generation of misaligned images. Consequently, the effectiveness of the model for end-users may be considerably compromised. **The robustness of T2I diffusion models is a crucial aspect for real-world deployment to ensure reliable performance across variations in inputs** [1]. Although this is not a security issue, it remains a critical factor influencing the practical utility of a diffusion model.
>
> **Question 2: How evaluating robustness can help improve user efficiency.**
>
> We are sorry for the potential confusion and make the following clarification. As discussed above, evaluating robustness can be used for model developers to evaluate the robustness of their T2I models before being deployed by simulating real-world input errors from humans to ensure utility for users. If the robustness has been improved during deployment phase, the final T2I models for users could not only improve user experience and enhance efficiency of use, but also reduce user costs by avoiding the need to go back and check mistakes in the prompts and make corrections after generating erroneous images.
>
> Further discussion is welcome if you have any other questions. Thank you again.
>
> **Reference**: [1] Lee et al., Holistic Evaluation of Text-to-Image Models, NeurIPS 2023 Track on Datasets and Benchmarks.

---

> > ### Comment · Reviewer_BYdd · 2023-11-22
> >
> > Thank you for your response.

---

### Official Review · Reviewer_st1Q · 2023-10-27

**Soundness:** 2 fair
**Presentation:** 3 good
**Contribution:** 3 good
**Rating:** 6
**Confidence:** 4

**Summary:**

This work introduces a method for finding minor modifications to text inputs in text-to-image generating models that induce errors in the generated image. Modifications take the form of typos, replacements with unicode characters, and replacements with similar-sounding letters. The errors the algorithm induces are images that are unfaithful to the prompt in some way. For example, one image is missing any depiction of ice cream when prompted for “A giant icae cream sculpture towering over a miniature town.” The method developed has three components. First, the most important words in the prompt are selected via word ablation. Then, random modifications are applied to that word (typos, etc.). Finally, the most successful modifications are adopted and the iteration repeats until a stopping condition is met. In order to instantiate this algorithm, the authors adapt and develop measurements for distance between distributions of texts and distributions of images.

In the parlance of the adversarial examples literature, this can be considered an untargeted black box attack.

**Strengths:**

The paper’s main contribution is in adapting metrics for distance between distributions to the problem of generating “adversarial examples” for text-to-image models. While the final algorithm is quite simple at its core, the choice of metrics to use for attacking models that have non-deterministic outputs is not immediately clear. This paper examines the design space and its conclusions are useful reading for the community.

It is also important to note that, to my knowledge, this is the first paper that strives to preserve the semantics of the input prompt in attacks on text-to-image generating models. Even though, as I point out in the Weaknesses section, how much those semantics are actually preserved is arguable, it is important that the method does not produce gibberish prompts.

**Weaknesses:**

The paper should better address the question of whether the perturbed text meaningfully alters the semantics of the text – to the point where the images produced remain faithful to the modified prompt, even if different from the intention of the original prompt. This has particular relevance since the paper presents an untargeted attack and untargeted attacks in image generation can have a very loose definition. This has three prongs:
1. The example prompt and image pairs that are selected in Figures 1, 6, 7, and 8 can reasonably be challenged on the grounds that many modifications, although minor in Levenstein distance, actually alter the semantics. For example, it is not obvious that the word “pamda” with an “m” should produce an image the animal panda.
2. It is unclear what the following quantities are precisely: human labeler-provided consistency scores (in particular, what are N1 and N2 in Appendix D?), SI2T, and ST2I.
3. The prompts that humans typed in seem rather to be rather easy attempts that do not alter the text semantics as much as the auto-modified prompts do. But humans can conceivably try harder. Is there a way to normalize for the strength of the attack?

Another point on which the paper can be improved is to develop methods for generating images based on the adversary’s chosen semantics. In many practical scenarios, it is important to understand if images can draw very specific violating concepts. For example, adversaries might want to generate specific public figures in compromising scenarios.

The authors might also improve the paper by including a discussion of the ability to undo the adversarial modifications with different tokenization, with hand-written rules, or with Large Language Models that undo typos and glyph and phonetic replacements.

**Questions:**

Can you define exactly how SI2T and ST2I and their adversarial counterparts are computed?

Can you define the human-provided consistency scores described in Appendix D (N1 and N2)? Why is a difference greater than 1 (as opposed to 0) the “pivotal” difference determining attack success rate?

---

> ### Author Response · Authors · 2023-11-17
> **Response to Reviewer st1Q(1/2)**
>
> Thank you for the valuable and comprehensive review. Below we address the detailed comments and hope that you may find our response satisfactory.
>
> **Question 1: The example prompt and image pairs that are selected can reasonably be challenged on the grounds that many modifications, although minor in Levenstein distance, actually alter the semantics.**
>
> We are sorry for the potential confusion and make the following clarification in two perspectives. On the one hand, for users, the mistakes made by themselves won't affect the semantics as perceived by the users because they did not notice these mistakes. On the other hand, since the diffusion models typically rely on context to generate images, minor perturbations usually do not impact the model's understanding of the semantic meaning of the sentence. As demonstrated in Figure 9 in the Appendix, the diffusion model can still generate realistic and aligned images despite perturbing many words in the text prompt. Therefore, for smaller perturbations like those in our case, diffusion models are expected to produce images that resemble the original ones. However, our perturbed prompts successfully induced diffusion models to generate mismatched images. Besides, other kinds of perturbations (glyph and phonetic) are more imperceptible but can lead to significantly different images, as shown in Figures 1, 6, and 7.
>
> **Question 2: Can you define exactly how SI2T and ST2I and their adversarial counterparts are computed?**
>
> As described in the Appendix B.1, $\text{$S_{I2T}$}= \frac{1}{N} \sum_{i=1}^N \max(0, 100\cdot g_{\phi}(c)^\top h_{\phi}(x))$ calculates the cosine similarity between the embeddings of images and text, no matter whether the images are generated by the original text or the adversarial text. $\text{$S_{T2T}$}= \max(0, 100\cdot g_{\phi}(c)^\top g_{\phi}(c'))$ calculates the cosine similarity between the embeddings of original text and the adversarial text.
>
> **Question 3: Can you define the human-provided consistency scores described in Appendix D (N1 and N2)?**
>
> The Description Consistency Rate (DCR) in Appendix D differs from $N_1$ and $N_2$ in Appendix B.2. The former evaluates the consistency between sentences, while the latter evaluates the consistency between sentence and image.
> As described in Appendix D.2, we sampled 100 texts from successfully attacked texts for each attack method and evaluated the description consistency between these adversarial texts and their corresponding original texts by humans. To avoid bias, we evaluated each text with three people and took the plural of the three people’s opinions as the final decision. This is how we calculate the Description Consistency Rate (DCR).
> As described in Appendix B.2, $N_1$ represents the consistency between the original text and the original image, and $N_2$ represents the consistency between the original text and the adversarial image, which only needs human annotators to evaluate directly.
>
> **Question 4: Why is a difference greater than 1 (as opposed to 0) the “pivotal” difference determining attack success rate?**
>
> Thank you for kindly pointing it out. Due to the inherent randomness in image generation, we believe that solely relying on $N_1$ not equal to $N_2$ (i.e., $N_1 - N_2 > 0$) is not a sufficiently stringent condition to define a successful attack. Therefore, we impose a more stringent criterion, requiring ($N_1 - N_2$) > 1 as a condition for a successful attack. Similarly, we consider $N_1 - N_2 < 1$ as the condition for filtering out meaningless prompts.
>
> **Question 5: Is there a way to normalize for the strength of the attack?**
>
> Thanks for the comment. Perturbation rate can be used for measuring the strength of an attack. Figure 3 illustrates the generation performance of diffusion models under various attack strengths. The horizontal axis represents the perturbation rate, serving as a measure of attack strength, where a higher perturbation rate indicates a stronger attack. The vertical axis depicts the consistency between the images generated by diffusion models and the corresponding text at different attack strengths, representing the overall generation effectiveness of diffusion models.

---

> ### Author Response · Authors · 2023-11-17
> **Response to Reviewer st1Q(2/2)**
>
> **Question 6: The paper can be improved by developing methods for generating images based on the adversary’s chosen semantics.**
>
> Thank you for your kind suggestion. Since our attacks simulate human mistakes in real-world scenarios, they are more effective for non-targeted attacks. In the introduction, we emphasized the distinction between our approach and malicious attacks, highlighting our focus on a more realistic and objective form of attack, devoid of subjective chosen targeted semantics. We will further study the application of our method to targeted attacks in future work.
>
> **Question 7: Improve the paper by including a discussion of the ability to undo the adversarial modifications with different tokenization, with hand-written rules, or with Large Language Models that undo typos and glyph and phonetic replacements.**
>
> Thank you for your thoughtful suggestions. Given that the targeted model is a black-box model, modifying tokenization is not within our control. Regarding the typo correction, as outlined in Appendix C.2 of our paper, we employed two typo correction tools with different rules. We found that some of our texts can be maintained as adversarial samples.
>
> Additionally, large language models are indeed effective in mitigating character-level attacks. We employed ChatGPT to repair samples generated by three attack methods: typo, glyph, and phonetic. With 50 samples for each method, the success rate of repairs exceeded 90%. We add this discussion in Appendix C.3.

---

> > ### Author Response · Authors · 2023-11-23
> > **Sincerely looking forward to the further discussions**
> >
> > Dear reviewer,
> >
> > We are wondering if our clarification of contributions and response have resolved your concerns. If our response has addressed your concerns, we hope for your reconsideration in raising your score.
> >
> > If you have any additional questions or suggestions, we would be happy to have further discussions.
> >
> > Best regards,
> >
> > The Authors

---

### Official Review · Reviewer_BaK3 · 2023-11-01

**Soundness:** 2 fair
**Presentation:** 2 fair
**Contribution:** 2 fair
**Rating:** 3
**Confidence:** 3

**Summary:**

This paper proposes to evaluate the robustness of the off-the-shelf text-to-image diffusion models r.w.t. the text perturbations including the typo, glyph, and phonetic. Different from other works that focus on textual adversarial attacks, this paper focuses on these text modifications that commonly happen in real-world applications. By proposing four different adversarial attack objectives, e.g., maximum mean discrepancy, and KL-divergence, this paper proposes to perturb the text condition to ensure these objectives. Experiments on several off-the-shelf T2I diffusion models show the attack objectives contribute to reducing the similarity between text and images.

**Strengths:**

1. This paper studies an interesting and important problem, i.e., the adversarial robustness of the text-to-image diffusion model. The text-to-image diffusion model has shown its power in generating images in the zero-shot setting but the robustness to the adversarial attacks is under-studied. This paper focuses on a significant problem to evaluate the robustness against the common perturbations.

2. The writing is easy to understand and the concepts are well illustrated with figures.

**Weaknesses:**

1. The main concern is the significance of the problem setting/definition. First of all, generally, the attack should be deployed with some malicious goal, while this paper considers the typos/glyph/phonetics as "real-world attacks" which could be controversial. In the real-world scenario, the existing typos/glyphs/phonetics are usually mistakenly caused by the users of T2I applications without malicious intentions, and even with a bad generation, the users can go back to check the mistakes in the prompts and fix them. It is hard to understand the purpose of these "adversarial attacks" in this scenario. Second, whether the robustness of the diffusion model to the text input is good or bad is another important question. In Figure 1, it is hard to say whether the generations based on these modified texts are "wrong". From another perspective, we can even conclude that these text-to-image models are "accurate" in generating images that reflect the text. Since the texts and their semantic meanings have changed, the generation "accurately" reflects the changes. It is also controversial that what criteria does the human evaluation adopts. Is it based on the difference between the image content and the original meaning of the text or the actual meaning of the text?

2. All the examples are based on the deviation of the objects in the texts. As one character change in the noun can lead to another meaning, it is normal to see that the T2I does not generate the correct objects. The question is does this "mismatch" only happen in the noun words?

3. The details of the human evaluation are not clear. Also, even with the clean inputs, the mismatch in the existing T2I models happens. The authors should provide the human evaluation results on the original text and corresponding images as a baseline, otherwise the human evaluation results in Table 2&3 are meaningless.

**Questions:**

1. Is the human evaluation based on the difference between the image content and the original meaning of the text or the actual meaning of the text?

2. Does the mismatch only happen on noun words?

3. What is the possible application of these real-world attacks?

4. Can you provide the human evaluation on the original task as the baseline?

**Details Of Ethics Concerns:**

This paper does adversarial attacks on the commercial applications of the Text-to-Image models, e.g., Stable Diffusion and DALL-E, as well as the human evaluation of the adversarial examples, which may require approval from the ethical consideration departments.

---

> ### Author Response · Authors · 2023-11-17
> **Response to Reviewer BaK3(1/2)**
>
> Thank you for the valuable and comprehensive review. Below we address the detailed comments. We hope that you may find our response satisfactory and could kindly raise your score.
>
>
> **Question 1: It is hard to understand the purpose of these "adversarial attacks" in this scenario.**
>
> We are sorry for the potential confusion and make the following clarification. As described in the introduction, the goal of this paper is to evaluate the robustness of text-to-image diffusion models under real-world perturbations. Such perturbations can stem from human mistakes and have the potential to induce misaligned images. Thus, the model's efficacy for users can be significantly diminished. The robustness of T2I diffusion models is a crucial aspect for real-world deployment to ensure reliable performance across variations in inputs [1].
>
> To evaluate model robustness, we adopt adversarial attacks as an indispensable method. Our attack method can be used by the model developers to evaluate the robustness of their T2I models before being deployed to ensure utility for users. In other words, the attacker is more likely to be the model developer rather than a malicious user. Robust T2I models not only enhance user efficiency but also reduce user costs by avoiding the need to go back and check mistakes in the prompts and make corrections after generating erroneous images. Overall, our work can be significant for studying the vulnerabilities of T2I models, as agreed by Reviewer BYdd.
>
> **Question 2: Whether the robustness of the diffusion model to the text input is good or bad is another important question.**
>
> Thanks for the comments and we make the following clarifications. We believe that the robustness of diffusion models to input variations is good and crucial for real-world deployment. As illustrated in Figure 9 in the Appendix, the diffusion model demonstrates an ability to generate realistic and aligned images even when most words in the image are randomly perturbed. This is a desirable ability of diffusion models because the generations can better conform to the user’s real intention instead of the potential bias in the provided prompts [1]. This work evaluates such an ability of diffusion models by adversarially perturbing the prompt and checking to what extent the mismatched generations occur.
>
> **Question 3: Is the human evaluation based on the difference between the image content and the original meaning of the text or the actual meaning of the text?**
>
> As we discussed above, generating an image that aligns with the original text should be deemed as 'good.' Therefore, human evaluation is based on evaluating the disparity between the image content and the original meaning of the text.
>
> To prove that human evaluation won’t be affected by the adversarial noise of text. We also added an experiment to evaluate the difference between the image content and the actual meaning of the adversarial text in revision. We let $N_1$ represents the number of images generated by the original text that are consistent with the original text, while $N_2$ represents the number of images generated by the adversarial text that are consistent with the **adversarial text**. The new overall human evaluation score is calculated in the same way as the original paper (also see Question 5 for details). Table 8 in Appendix D.3 shows the new overall human evaluation on the same adversarial dataset generated under real-world attack with the 2ST attack objective in Section 5. We found that although the new overall human evaluation score will decrease to some extent compared to the old score, the change is not significant, **indicating that human evaluations are not affected by the adversarial text**. We also show this table below：
>
> |Dataset |Attacker |$Ori.S_{I2T} $|Ave. Len.| L-distance |Old Hum.Eval   | New Hum. Eval|
> | :--: | :--: | :--: |  :--: | :--: | :--: |   :--: |
> |	|Typo|		|	|2.92                | 84.34% |82.19%|
> |ChatGPT-GP|Glyph| 27.61±2.07|10.41|2.27|84.65%|83.42%|
> |	|Phonetic|	|	|5.38|	86.16%|82.06%|
> |	|Typo|		|	|2.29                | 76.64% |74.34%|
> |DiffusionDB|Glyph| 29.17±3.36|10.62|1.81|76.64%|75.26%|
> |	|Phonetic|	|	|5.04|	75.51%|73.61%|
> |	|Typo|		|	|2.08               | 80.21% |78.28%|
> |LAION-COCO|Glyph| 27.54±2.86|9.17|1.85|81.89%|80.94%|
> |	|Phonetic|	|	|5.04|	79.32%|77.02%|
> |	|Typo|		|	|2.97             | 84.34% |82.13%|
> |SBU Corpus|Glyph| 24.99±3.43|11.69|2.42|85.41%|85.57%|
> |	|Phonetic|	|	|5.85|	85.41%|81.95%|

---

> ### Author Response · Authors · 2023-11-17
> **Response to Reviewer BaK3(2/2)**
>
> **Question 4: Does this "mismatch" only happen in the noun words?**
>
> This ”mismatch“ typically occurs in nouns, adjectives, and verbs. Since the cases we provide are descriptive sentences about objects, there are more instances of mismatches in nouns. It is important to note that our approach aims to identify words that are most important for the model, rather than those deemed most important by humans. Therefore, in theory, it is not limited to a specific part of speech. We add some cases without noun modification in Figure 10 in Appendix D.4.
>
> **Question 5: The details of the human evaluation are not clear. Can you provide the human evaluation on the original task as the baseline?**
>
> We agree that the mismatch also exists for clean inputs. We deal with this problem by measuring the percentage of samples that leads to mismatch while the clean inputs do not. As described in Appendix B.2,  $N_1$ represents the number of images generated by the original text that are consistent with the original text, and $N_2$ represents the number of images generated by the adversarial text that are consistent with original text. If ($N_2 - N_1$) > 1, the attack on that particular prompt text is deemed meaningless. Let's assume the frequency of samples where ($N_2 - N_1$) > 1 as $N_u$, and the effective total number of samples should be $N_{\text{total}} - N_u$. If $(N_1 - N_2 > 1)$, it indicates a successful attack. We use $N_c$ to represent the number of samples where the attack is successful. Thus, the final score for each evaluator is given by $N_c / (N_{\text{total}} - N_u)$. The average of three human annotators represents the overall human evaluation score.
>
> **Question 6: What is the possible application of these real-world attacks?**
>
> As we discussed in Question 1, the goal of this paper is to evaluate the robustness of text-to-image diffusion models under real-world perturbations. To evaluate model robustness, we adopt adversarial attacks as an indispensable method. Therefore, **the possible application of our proposed real-world attack approach is for model developers to evaluate the robustness of their T2I models before being deployed by simulating real-world input errors from humans to ensure utility for users.**
>
> **Reference:**
> [1] Lee et al., Holistic Evaluation of Text-to-Image Models, NeurIPS 2023 Track on Datasets and Benchmarks.

---

> > ### Author Response · Authors · 2023-11-23
> > **Sincerely looking forward to the further discussions**
> >
> > Dear reviewer,
> >
> > We are wondering if our clarification of contributions and response have resolved your concerns. If our response has addressed your concerns, we hope for your reconsideration in raising your score.
> >
> > If you have any additional questions or suggestions, we would be happy to have further discussions.
> >
> > Best regards,
> >
> > The Authors

---

> > > ### Comment · Reviewer_BaK3 · 2023-11-23
> > >
> > > Thanks for the response. The comments do not convince me of the motivation of the attack and the human evaluation is still ambiguous. The model developer may want the model to be less robust rather than more robust. It is hard to say whether making it more robust is good or making it more sensitive is good. The robustness against these attacks (typos) can lead to less accurate generation and limit the diversity of the generation. The human evaluation, when evaluating whether the generation is consistent with the adversarial text, should mark the images as wrong if they capture the original meaning of the adversarial text, e.g., an image of a pear tree is a wrong generation for "pearl tree" even if the "pear" is mistakenly typed as "pearl".
> > >
> > > In conclusion, different from the adversarial robustness in the classification task where "correct" and "incorrect" are clearly defined,  the adversarial robustness in the text-to-image task is not clear in this paper since the evaluation of the generation is sometimes subjective, and even opposite. Without a clear definition of the adversarial robustness in this task, the motivation and the evaluation would be controversial.

---

> ### Author Response · Authors · 2023-11-23
> **Further clarification**
>
> Thank you for your response. I believe there might be some misunderstanding here. The attribute you mentioned, which makes the model more responsive to inputs, is commonly referred to as **model sensitivity**[1]. It is a concept opposite to **robustness**. Our focus here is primarily on exploring model robustness, which is typically defined as the ability to maintain the original output after applying imperceptible perturbations[2,3], as opposed to sensitivity, which involves generating different outputs in response to perturbations.
>
> Balancing the exploration of robustness and sensitivity is fundamentally an important topic[1]. However, in our work, we are specifically considering robustness, which is clearly and solely defined as the capability to preserve the original output under external perturbations. Therefore, there is no ambiguity in the definition.
>
> **References:**
>
> [1] Kim et al. Balancing Robustness and Sensitivity using Feature Contrastive Learning.  arXiv:2105.09394 (2021).
>
> [2] Chen et al. From Adversarial Arms Race to Model-centric Evaluation: Motivating a Unified Automatic Robustness Evaluation Framework. Findings of ACL 2023.
>
> [3] Lee et al., Holistic Evaluation of Text-to-Image Models, NeurIPS 2023 Track on Datasets and Benchmarks.

---

### Official Review · Reviewer_a3Ko · 2023-11-01

**Soundness:** 3 good
**Presentation:** 3 good
**Contribution:** 3 good
**Rating:** 5
**Confidence:** 4

**Summary:**

This paper evaluate the robustness of T2I DMs by adversarial attacks. Specifically, this paper treats realistic errors (e.g., typo, glyph, phonetic) that humans can make as minor perturbations and optimizes adversarial perturbations under distribution guidance. The method is validated on stable diffusion and DALL-E 2.

**Strengths:**

-- Robustness evaluation of T2I DMs is an important and interesting problem; this paper reveals that small perturbations to the prompt can result in significant changes in the generated images.

-- Realistic errors are used to attack DMs. This perturbation is more subtle and is more likely to occur in real-world scenarios.

-- Distribution guidance is used to optimize adversarial samples. Compared to targeting a single image, this approach is more novel and has been proven to be more effective by the authors.

**Weaknesses:**

This paper focus on an important and interesting problem, i.e., robustness evaluation of T2I DMs, and proposes a novel distribution guidance method for generating adversarial samples. However, its performance is not fully validated, and the authors need to provide enough details to guarantee its reproducibility.

-- The insight of the definition of robustness in the paper is that, after minor perturbation perturbations to the prompt, DMs should still generate similar images. However, based on the examples provided in the paper, successful attacks have led to changes in the subject of prompts (e.g., in Fig.7, ‘panda’ changed to ‘pamda’), and images generated from such prompts should inherently be different. The authors need to demonstrate the generation of significantly different images when the perturbed prompt and the original prompt should, in fact, result in similar images.

-- The authors mentioned that introducing distribution guidance is aimed at addressing the randomness. However, they only compare the performance of distribution guidance and single image guidance in terms of the decrease in distribution metrics. To better demonstrate the ability of distribution to handle randomness, the authors should include experimental results for single image guidance in the 'real-world attack experiment' section.

**Questions:**

Please see the two questions list in the weaknesses part.

---

> ### Author Response · Authors · 2023-11-17
> **Response to Reviewer a3Ko**
>
> Thank you for the valuable review. Below we address the detailed comments and hope that you may find our response satisfactory.
>
> **Question 1: The authors need to provide enough details to guarantee its reproducibility.**
>
> Thanks for the comments and we make the following clarifications. Firstly, Section 3.3 contains detailed information about our method. In addition, the code we submitted includes sufficient details to ensure the reproducibility of our method, as agreed by Reviewer BYdd.
>
> **Question 2: Based on the examples provided in the paper, successful attacks have led to changes in the subject of prompts. The authors need to demonstrate the generation of significantly different images when the perturbed prompt and the original prompt should result in similar images.**
>
> We appreciate the comments and would like to provide the following clarifications. As illustrated in Figure 9 in the Appendix, the diffusion model has good robustness to random text perturbations, i.e., it can generate realistic and aligned images even when most words in the text prompt are perturbed. As a comparison, our method only modifies one or two words in the prompt to generate semantically different images. Although the typo may lead to nonsense words (e.g., “pamda”), they are realistic errors of humans. The other kinds of perturbations (glyph and phonetic) are more imperceptible but can lead to significantly different images, as shown in Figures 1, 6, and 7.
>
> **Question 3: The authors should include experimental results for single image guidance in the 'real-world attack experiment' section.**
>
> Thank you for the kind suggestion. We are delighted to add the experimental results for single-image guidance in the real-world attack experiment in the revision. Due to the extensive workload in this experiment, we have temporarily only added results for single image guidance on the ChatGPT-GP dataset in the real-world attack experiment with the 2ST attack objective. Once other results have been obtained, we will update the results to the revision as soon as possible. Table 5 in Appendix  C.3 shows that the $Adv. S_{I2T}$ and human evaluation scores associated with this objective are relatively worse. This observation suggests that the attack with the DI objective(i.e. single-image guidance attack)  may be susceptible to overfitting on a single image. We also show this table below：
>
>
> |Dataset |Attacker |$Ori.S_{I2T}$ |Ave. Len.| L-distance |$Adv. S_{I2T}  $  |Ave. Query  |Hum. Eval|
> | :--: | :--: | :--: |  :--: | :--: | :--: |   :--: |  :--: |
> |	|Typo|		|	|2.65                | 24.61±3.32               |16.97     |79.36%  |
> |ChatGPT-GP|Glyph| 27.61±2.07|10.41|2.09|24.94±3.01|16.86|77.45%|
> |	|Phonetic|	|	|5.13|	26.67±3.74|15.71|78.68%|

---

> > ### Author Response · Authors · 2023-11-23
> > **Sincerely looking forward to the further discussions**
> >
> > Dear reviewer,
> >
> > We are wondering if our clarification of contributions and response have resolved your concerns. If our response has addressed your concerns, we hope for your reconsideration in raising your score.
> >
> > If you have any additional questions or suggestions, we would be happy to have further discussions.
> >
> > Best regards,
> >
> > The Authors

---

### Author Response · Authors · 2023-11-21
**Look forward to further feedback**

Dear reviewers,

We thank you again for the valuable and constructive comments. We are looking forward to hearing from you about any further feedback.

If you find our response satisfactory, we hope you might view this as a sufficient reason to further raise your rating.

If you still have questions about our paper, we are willing to answer them and improve our paper.

Best, Authors

---

### Meta-Review · Area_Chair_93uL · 2023-12-08

**Metareview:**

This paper studies the robustness of text-to-image models to minor perturbations in the prompt. The authors defined several perturbation types, e.g. typo, glyph substitution, and optimized over this space to find the perturbation that negatively influences the generated images the most. Empirical evaluation on stable diffusion and DALL-E 2 showed that these models are not robust to such perturbations, and can generate images that are semantically irrelevant to the text prompt according to both CLIP score and human evaluation.

Reviewer recommendations are split on this paper, with the most major weaknesses being its significance and practical value. Some of the perturbations are in fact not semantics-preserving, e.g. panda -> pamda, and it's arguable if the model's behavior is expected or not. There are also concerns about whether the paper's setting poses real security threat or not, since a benign user can simply correct their error in the prompt upon observing the generated image. Ultimately, the two reviewers recommending acceptance did not find the paper compelling enough to stand by their decision, and AC recommends rejection based on these two weaknesses.

**Justification For Why Not Higher Score:**

The two weaknesses are critical enough to warrant rejection. The two positive reviewers also did not championing its acceptance.

**Justification For Why Not Lower Score:**

N/A

---

### Decision · Program_Chairs · 2024-01-16

Reject